# Silicon and oxygen synergistic effects for the discovery of new high-performance nonfullerene acceptors

Ying Qin[1,6], Hui Chen[2,6], Jia Yao[3], Yue Zhou[1], Yongjoon Cho[4], Yulin Zhu[2], Beibei Qiu[5], Cheng-Wei Ju[1], Zhi-Guo Zhang[3✉], Feng He[2✉], Changduk Yang[4], Yongfang Li[5] & Dongbing Zhao[1✉]

In organic electronics, an aromatic fused ring is a basic unit that provides π-electrons to construct semiconductors and governs the device performance. The main challenge in developing new π-skeletons for tuning the material properties is the limitation of the available chemical approach. Herein, we successfully synthesize two pentacyclic siloxy-bridged π-conjugated isomers to investigate the synergistic effects of Si and O atoms on the geometric and electronic influence of π-units in organic electronics. Notably, the synthesis routes for both isomers possess several advantages over the previous approaches for delivering conventional aromatic fused-rings, such as environmentally benign tin-free synthesis and few synthetic steps. To explore their potential application as photovoltaic materials, two isomeric acceptor–donor–acceptor type acceptors based on these two isomers were developed, showing a decent device efficiency of 10%, which indicates the great potential of this SiO-bridged ladder-type unit for the development of new high-performance semiconductor materials.

[1] State Key Laboratory and Institute of Elemento-Organic Chemistry, College of Chemistry, Nankai University, Tianjin 300071, China. [2] Shenzhen Grubbs Institute and Department of Chemistry, Southern University of Science and Technology, Shenzhen 518055, China. [3] State Key Laboratory of Chemical Resource Engineering, Beijing Advanced Innovation Center for Soft Matter Science and Engineering, Beijing University of Chemical Technology, Beijing 100029, China. [4] Department of Energy Engineering, School of Energy and Chemical Engineering, Perovtronics Research Center, Low Dimensional Carbon Materials Center, Ulsan National Institute of Science and Technology (UNIST), Ulsan 44919, Republic of Korea. [5] Beijing National Laboratory for Molecular Sciences, CAS Key Laboratory of Organic Solids, Institute of Chemistry, Chinese Academy of Sciences, Beijing 100190, China. [6] These authors contributed equally: Ying Qin, Hui Chen. ✉email: zgzhang@mail.buct.edu.cn; hef@sustech.edu.cn; dongbing.chem@nankai.edu.cn

In comparison with its inorganic counterpart, such as classical silicon, organic electronics can promise unique features of solution processability, low weight, and flexibility[1–9]. These attractive advantages have inspired tremendous efforts in exploring the polycyclic aromatic hydrocarbons that can provide π-electrons to construct organic semiconductors, thus governing the device performance in a wide range of emerging technologies[10–15], such as organic light-emitting diodes (OLEDs), organic thin-film transistors (OFTs), and organic solar cells (OSCs). Although organic electronics have the highest market potential, their performances have not reached the practical requirement for broad commercial applications, mainly due to the suboptimal photophysical and electronic properties of these organic semiconductors. Fortunately, the properties of organic materials, including energy level, absorption, crystallinity, mobility, and electron affinity and stability in ambient air, can be flexibly adjusted by changing their chemical composition, such as the innovation of fused π-electron systems for obtaining high-performance polymeric and organic electronic materials[16–19]. In these bridged π-skeletons, the structural coplanarity and rigidity can reduce the conformational disorder, facilitate π-electron delocalization, and ultimately tune the photophysical and electronic properties.

In this context, the incorporation of silicon or oxygen atoms into the π-fused framework as bridging moieties has become a fruitful strategy to produce new π-materials possessing superb photophysical and electronic properties[20–25], which are difficult to achieve in carbon-based π-units. The longer C−Si bond length (C−Si vs. C−C: ~1.87 Å vs. ~1.53 Å), higher electropositivity (Si vs. C: 1.74 vs. 2.50), larger atomic radius (Si vs. C: 111 pm vs. 67 pm) and the particular interactions (σ*-π* conjugation) between the Si atom and the π-electron system often led to variation in energy levels, increased crystallinity, improved packing ability and higher charge mobility in comparison to those carbon-bridged π-conjugated scaffolds (Fig. 1a)[26–29]. Indeed, diverse 5-membered silicon-bridged π-systems, such as siloles, dibenzosiloles, dithienosiloles, and related compounds, have been synthesized and widely utilized in organic electronics (Fig. 1b)[29–41]. For example, Yang has shown that replacing carbon bridges with silicon bridges in cyclopentadithiophene-based polymers significantly improved the photovoltaic performance of polymer: PCBM-based solar cells[42]. On the other hand, recent studies revealed that the incorporation of electron-rich oxygen atoms into the bridge of ladder-type units would greatly improve the light-harvesting capability and the electron-donating capability of the unit, thus bringing unique characteristics to those new organic semiconductor materials to enhance the power conversion efficiency of organic photovoltaics[43]. Taking advantage of silicon and oxygen atom doping in the design of high-performance organic semiconductors, the development of 6-membered silicon-oxygen-bridged (SiO-bridged) π-conjugated skeletons to investigate the synergistic effects of Si and O atoms in organic electronics is highly appealing and would significantly enhance the structural diversity of π-conjugated materials and improve the performance in a wide range of optoelectronic technologies. However, until now, investigations on 6-membered SiO-bridged π-conjugated materials have been inherently limited by their narrow structural diversity, presumably mainly because of the challenges in their synthesis.

Herein, we designed two isomers of 6-memebered SiO-bridged ladder π-skeletons with different positions of silicon atoms, as shown in Fig. 1c. The silicon atom was connected to two thiophenes in SiO5T-5; in contrast, it was connected to the central benzene in SiO5T-10. This design aims to produce new donor units for the further development of high-performance organic/polymeric semiconductors. Several aspects of these two skeletons

are of interest. First, the thermal stability of siloxy-bridged ladder π-skeletons is expected to be higher because of the much larger average bond dissociation energy of the Si−O bond. Moreover, according to density functional theory (DFT) calculations on the molecular orbital using Gaussian 09 at the B3LYP/6-31 G(d) level, the Si−O bridge significantly alters the electronic structure of their C−O analogs. As shown in Fig. 1d, the 6-memebered SiO-bridged structures significantly enhance π-system planarity (10.9° vs. 0°; 8.5° vs. 3.8°) compared with the CO-bridged analogs. The enhanced planar configuration would be beneficial for charge transport properties. Furthermore, the prediction of the energy levels by the DFT calculation showed that the SiO-bridged SiO5T-5 and SiO5T-10 possess larger band gaps with lower-lying HOMO energy levels ($E_{HOMO}$) than their carbon analogs. For the two siloxy-bridged π-skeletons of SiO5T-5 and SiO5T-10, the calculated $E_{LUMO}/E_{HOMO}$ of SiO5T-5 is −1.52/−5.07 eV, which is significantly lower than that of the isomer SiO5T-10 ($E_{LUMO}/E_{HOMO}$ = −1.34/−4.98 eV), indicating the stronger interaction of the σ* orbital of the Si atom with the π* orbital of the π-skeleton in SiO5T-5. However, we may face some formidable challenges in our efforts to access the two siloxy-bridged ladder-type π-systems: (1) retro-Brook rearrangement may frequently occur if containing the siloxy-bridge in starting materials; (2) the instability of thiophenols is expected to lead to difficulty in the preparation of siloxy-bridged thiophenes. In this study, we addressed these problems in chemical synthesis and successfully accessed the two pentacyclic siloxy-bridged π-conjugated isomers SiO5Ts for the first time. Furthermore, we demonstrated that the incorporation of a SiO bridge into π-conjugated frameworks makes it possible to precisely tune the optoelectronic properties and molecular packing of π-systems, resulting in a decent device efficiency of 10% in polymer solar cells (PSCs) by the development of two novel isomeric siloxy-bridged A–D–A acceptors, which consist of our developed 6-membered laddered-SiO5T skeleton as the central donor unit and a dicyanomethylene derivative as the acceptor end group. This work is expected to enhance the discovery of high-performance materials for organic electronics using the 6-membered siloxy-bridged π- skeleton as the central core.

## Results

**Skeleton synthesis.** As mentioned above, the two ladder-type π-skeleton SiO5Ts have never been synthesized before. Herein, we tried to make a synthetic blueprint providing concise and rapid access to both isomers. In principle, the dibenzooxasiline moiety of the two ladder-type skeletons can be synthesized by Pd-catalyzed C−H bond arylation[44], Rh-catalyzed transmetalation[45] and Rh-catalyzed intramolecular C−H bond silylation[46]. However, both transmetalation and C−H bond silylation involved the corresponding phenols as key intermediates. Considering the instability of thiophenol, we envisioned that Pd-catalyzed intramolecular arylation of the C−H bond might be the only logical way to deliver the two siloxy-bridged ladder-type π-skeletons designed by us; even the competitive retro-Brook rearrangement can also occur during the C−H activation step[47]. We then carried out the synthesis of the two SiO5T isomers by intramolecular C−H bond arylation cyclization, as shown in Fig. 2. The lithium halogen exchange reaction of 3-bromothiophene 1 with $^{n}$BuLi followed by treatment with dichlorodialkylsilane yielded chlorodialkyl(thiophen-yl)silane 2, which further reacted with 2,5-dibromohydroquinone 3 to deliver the corresponding bissilyl ether 4, followed by subsequent Pd-catalyzed C−H bond arylation to give double-cyclized SiO-bridged SiO5T-5 (Fig. 2a). Following this success, we proceeded to synthesize the isomer of SiO-bridged SiO5T-5. After numerous attempts, we found that the

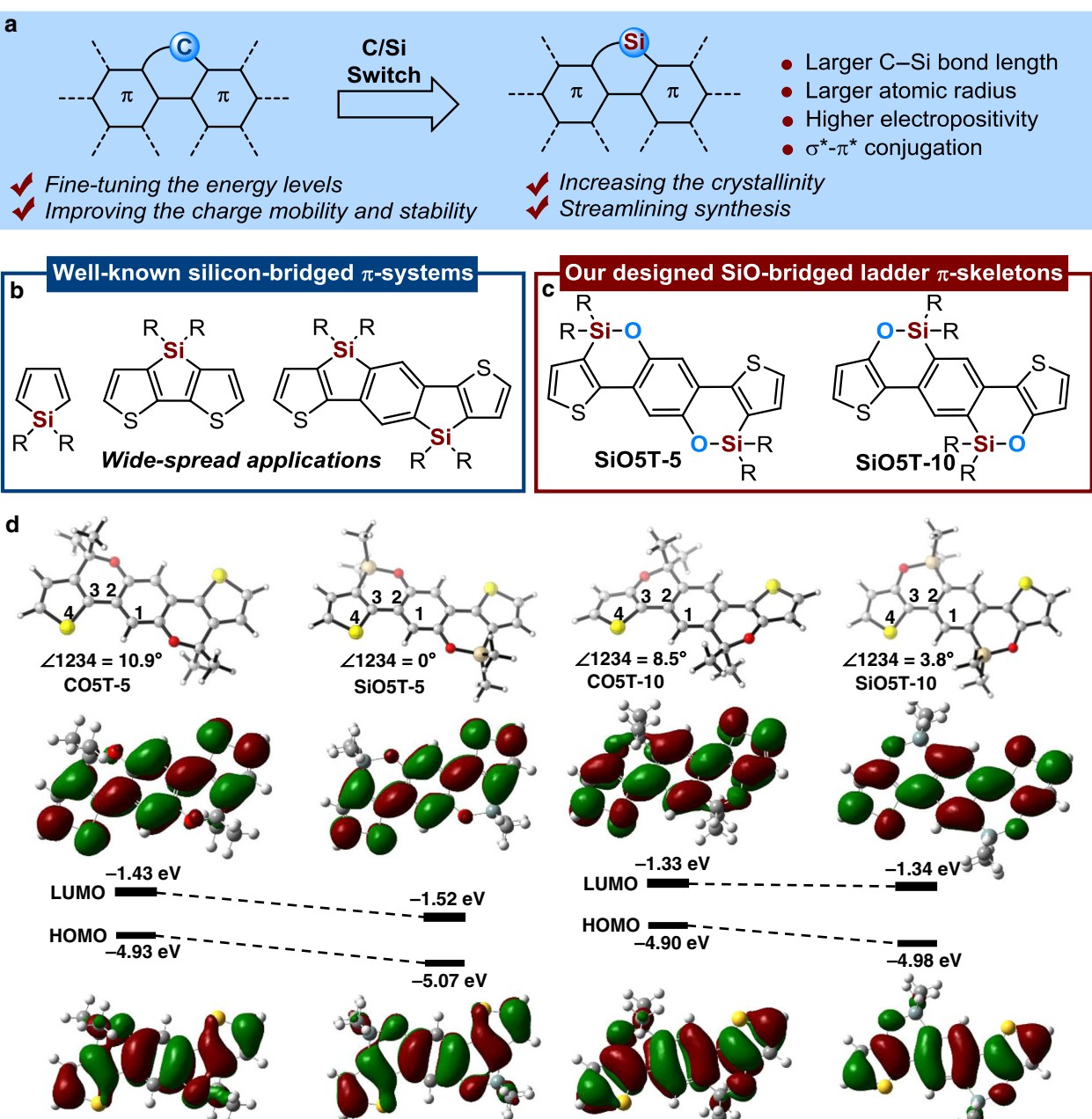

**Fig. 1 Design and development of 6-membered SiO-bridged ladder-type units. a** The advantage of incorporating silicon into π-systems. **b** Well-known silicon-bridged π-systems. **c** Our design on 6-membered SiO-bridged ladder-type units. **d** The molecular geometries and electron distributions of frontier orbitals of **SiO5Ts** and their carbon-analogs **CO5Ts**.

isomer **SiO5T-10** can also be easily approached in a four-step synthesis from 1,4-dibromobenzene (Fig. 2b). The C2 and C5-positions of the 1,4-dibromobenzene were lithiated by LDA and subsequently quenched with chlorodialkylhydrosilane, yielding bis-monohydrosilane **7** as a colorless oil. Further silane chlorination with TCCA and then reaction with 3-thiophenol in the presence of DMAP and imidazole in DCM afforded bissilyl ether **9** in 52% yield. Finally, the isomer **SiO5T-10** was obtained through a Pd-catalyzed intramolecular cyclization of aromatic C−H bonds. In general, the synthetic routes for both **SiO5T-5** and **SiO5T-10** possess several advantages over the approaches for delivering the corresponding C or CO-bridged IDTs and CO5Ts, such as low cost, tin-free synthesis, and few synthetic steps.

**Development of SiO-bridged acceptor materials.** The development of new *n*-type small-molecule semiconductor materials as electron acceptors in bulk heterojunction (BHJ) organic photovoltaics (OPVs) has progressed rapidly and attracted worldwide attention in recent years because of the advantages of these materials over typical fullerene derivatives in terms of structural flexibility and absorbent tunability, thermal/photochemical stability, solubility/processability and synthetic costs[48–52]. Among the reported nonfullerene acceptors, acceptor–donor–acceptor (A-D-A) nonfullerene acceptors based on ladder-type π-systems seem to be the most successful to date due to their high electron mobility, suitable electronic energy levels, low bandgap, broad and strong absorpiton[53–56]. The further development of a novel ladder-type backbone is the core to construct new A-D-A

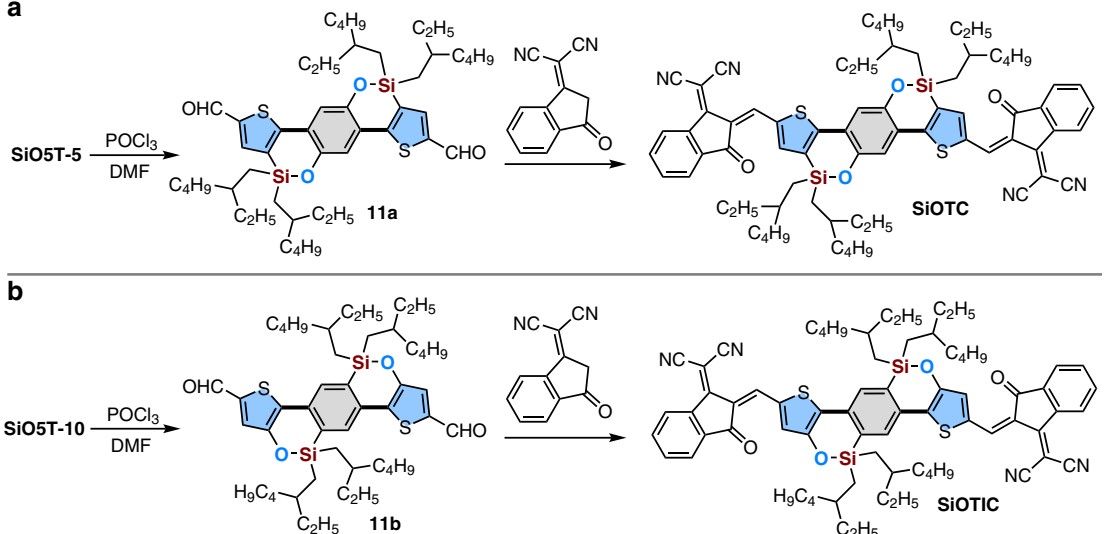

**Fig. 2 Synthetic routes for the SiO-bridged ladder-type π-skeletons. a** The route for producing ladder-type π-skeleton **SiO5T-5. b** The route for producing ladder-type π-skeleton **SiO5T-10**.

**Fig. 3 Development of the two A-D-A nonfullerene acceptors based on ladder-type SiO-bridged π-systems. a** The synthetic route for the acceptor **SiOTC. b** The synthetic route for the acceptor **SiOTIC**.

nonfullerene acceptors for overcoming the issues in OPVs. Inspired by the good device performance obtained from those IDT-[57–60] and CO5T-based[61–65] ladder-type A-D-A nonfullerene acceptors, we speculated that the development of new A-D-A nonfullerene acceptors based on SiO-bridged ladder-type π-skeleton SiO5Ts can be a fruitful avenue for further improving the performance of nonfullerene PSCs because of silicon and oxygen synergistic effects. Herein, two isomers of A-D-A nonfullerene acceptors (**SiOTC** and **SiOTIC**) were designed and synthesized, as shown in Fig. 3, by bis-formylation of the C2-position of thiophene units at **SiO5T-5** or **10** to deliver the corresponding aldehyde, followed by condensation with the end group 2-(3-oxo-2,3-dihydroinden-1-ylidene)malononitrile (IC).

The optical properties of **SiOTC** and **SiOTIC** in dichloromethane (DCM) solution and in thin films were measured by

UV-vis absorption spectroscopy, as shown in Fig. 4a, b. The corresponding spectroscopic data are summarized in Table 1. These two isomers displayed very similar absorption profiles in solution with two peaks, which can be assigned as the (0-1) and (0-0) vibronic transitions, respectively (Fig. 4a). **SiOTIC** presented redshifted absorption relative to that of **SiOTC** in both solution and solid films. This is believed to be associated with the different electron-withdrawing effects of the silicon atoms in the two isomers. Moreover, the **SiOTIC** acceptor also exhibited a higher maximum extinction coefficient with red-shifted absorption relative to that of **SiOTC** in DCM (Fig. 4b; $1.27 \times 10^5$ vs. $0.99 \times 10^5$ M$^{-1}$ cm$^{-1}$), which would benefit harvesting solar light and enhancing $J_{SC}$. Both isomer films presented broader absorption ranges (500–800 nm) and redshifted maximum absorption peaks (699 nm for **SiOTC** and 717 nm for

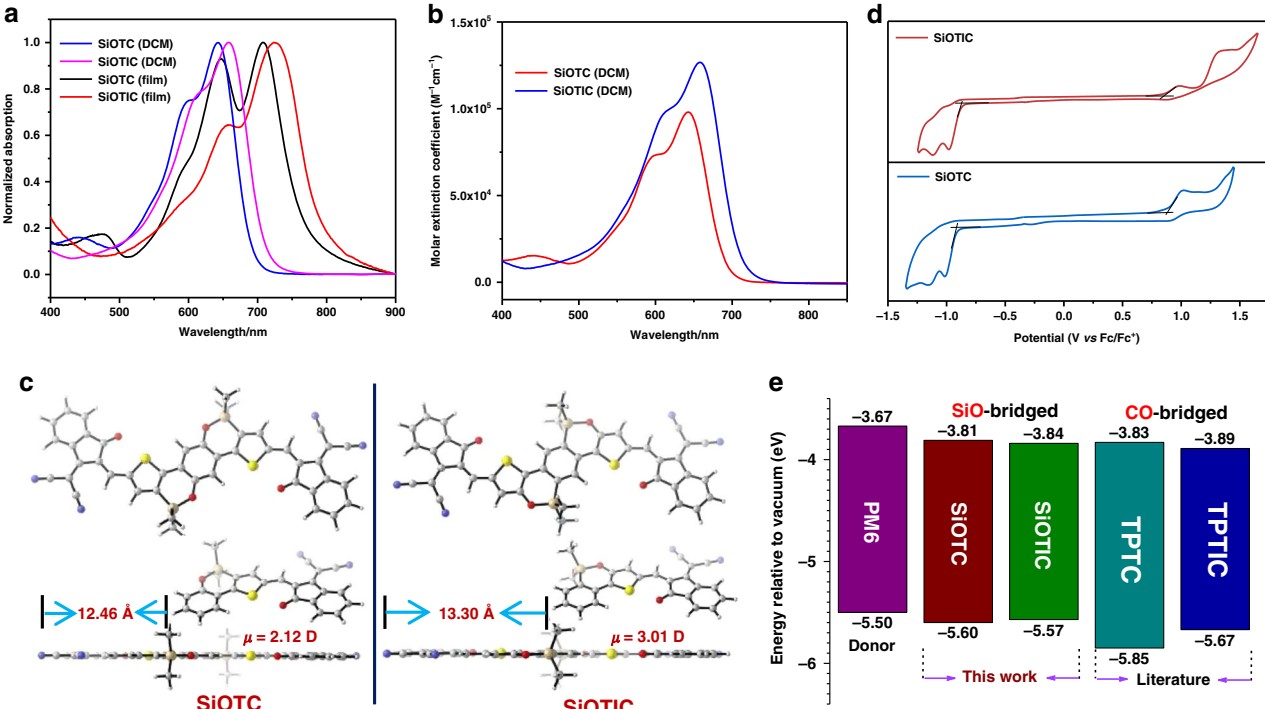

**Fig. 4 Molecular physicochemical and optical properties of the acceptors. a** UV–vis absorption of **SiOTC** and **SiOTIC** in DCM and film. **b** Comparison of the molar extinction coefficient between **SiOTC** and **SiOTIC** in DCM. **c** Optimized molecular geometries of **SiOTC** and **SiOTIC** by DFT calculations. **d** Cyclic voltammograms of **SiOTC** and **SiOTIC** in solution. **e** Energy level diagram of the donor and acceptor photovoltaic materials.

**SiOTIC**), indicating that the films of the two isomers showed stronger π–π stacking interactions and more ordered aggregation. Furthermore, the 0-1 absorption peak of the **SiOTC** acceptor is significantly strengthened in the solid film, indicating the stronger aggregation of **SiOTC**, which may lead to a complicated aggregation morphology and could be harmful to electron separation and transportation in the organic photovoltaic device. In contrast, the 0–1 absorption peak of the **SiOTIC** acceptor in the film is decreased in comparison with that of its solution. We rationalized that the more compact alkyl chains of **SiOTIC** may prevent severe aggregation, which would benefit the solution processability in device fabrication. The optical bandgaps of the two acceptors were also estimated from the absorption edge, as shown in Table 1. The optical bandgap of the **SiOTIC** film is smaller than that of **SiOTC** (1.55 *vs.* 1.61 eV), which would benefit enhancing $J_{SC}$ and harvesting low-energy sunlight.

DFT calculations were carried out to understand the essential differences in the backbone conformation and dipole moment of these two isomers. The side chains of the two isomers were simplified to methyl groups for the simulation. As shown in Fig. 4c, both **SiOTC** and **SiOTIC** possessed highly coplanar conformations. The main difference between **SiOTC** and **SiOTIC** is from the steric position of the alkyl groups on the silicon atom in the molecular geometries. SiOTIC showed a longer distance of 13.30 Å between the center of the side chain and the edge of the IC end group than **SiOTC** (12.46 Å), which allows more space for intermolecular antiparallel stacking of the terminal IC moieties. The dipole moments of the half **SiOTC** and **SiOTIC** molecules were also estimated to be 2.12 D and 3.01 D, respectively. The larger dipole moment of the half **SiOTIC** molecule is responsible for the stronger intramolecular charge transfer and redshifted absorption spectra.

To further study the electronic energy level and bandgap of the two isomers, electrochemical cyclic voltammetry (CV) was used to probe their redox properties, as shown in Fig. 4d. In terms of the onset reduction and oxidation potentials, the LUMO/HOMO energy levels of **SiOTC** and **SiOTIC** were estimated to be −3.81/ −5.60 eV (1.79 eV) and −3.84/−5.57 eV (1.73 eV), the trend of which is in agreement with that obtained from DFT calculations. **SiOTIC** has an up-shifted HOMO compared to that of **SiOTC**, which could be attributed to the position effects of silicon atoms in the isomeric acceptors. As shown in Fig. 4e and Supplementary Figs. 1 and 2, the SiO-bridged acceptors **SiOTC** and **SiOTIC** possess high-lying energy levels for both the LUMO and HOMO relative to those of C-analog **IDIC**[58, 59] and CO-analogs **TPTC** and **TPTIC**[65]. The DFT calculations revealed that the better planarity and an enhanced push-pull effect between the electron donor/acceptor units on **SiOTC/SiOTIC**, as shown in Supplementary Figs. 3 and 4, is responsible for these changes in the energy level. This result suggested that the SiO-bridged acceptors have the potential to deliver a higher open-circuit voltage ($V_{oc}$) in a polymer solar cell (PSC) device than those of the C- and CO-bridged analog-based devices. Both thermogravimetric analysis (TGA) and differential scanning calorimetry (DSC) were employed to investigate the thermal stability of these two isomers (Supplementary Fig. 5). Thermogravimetric analysis (TGA) revealed that both isomers possess fairly good thermal stability with thermal decomposition temperatures ($T_d$) of 5% weight loss at 339 °C for **SiOTC** and 323 °C for **SiOTIC**. The existence of the obvious melting peak in the first heating of the DSC testing revealed that there are crystallite structures in both isomers. Notably, **SiOTIC** shows a higher melting temperature (289.5 °C) than **SiOTC** (276.3 °C), indicating that **SiOTIC** may undergo more compact stacking in the film, which is harder to disrupt in device fabrication; thus, thermal annealing might be needed. In general, all these studies of optical properties, electronic energy levels, and crystallinity proved that the incorporation of silicon and oxygen atoms and the isomerization of the fused-ring central

**Table 1 Optoelectronic properties of the two SiO-bridged A–D–A nonfullerene acceptors.**

| Comp. | Solution[a] | | | Film | | Energy level (DFT) | | Energy level (CV)[c] | | $E_g^{opt\ d}$ | $E_g^{cv\ e}$ |
|---|---|---|---|---|---|---|---|---|---|---|---|
| | $\lambda_{max}$ [nm] | $\lambda_{onset}$ [nm] | $\varepsilon_{max}^{b}$ [M$^{-1}$ cm$^{-1}$] | $\lambda_{max}$ [nm] | $\lambda_{onset}$ [nm] | LUMO (eV) | HOMO (eV) | LUMO (eV) | HOMO (eV) | | |
| SiOTC | 642 | 699 | $0.99 \times 10^5$ | 708 | 772 | −3.47 | −5.67 | −3.81 | −5.60 | 1.61 | 1.79 |
| SiOTIC | 658 | 717 | $1.27 \times 10^5$ | 724 | 800 | −3.49 | −5.64 | −3.84 | −5.57 | 1.55 | 1.73 |

[a]DCM was used as the solvent; [b]Calculated at $\lambda_{max}$; [c]Calculated according to the equation $E_{LUMO/HOMO} = -e(E_{red/ox} + 4.38)$ (eV); [d]Calculated from the onset wavelength of the molecules in film: $E_g^{opt} = 1240/\lambda_{onset}$; [e]Calculated from the CV diagram in DCM solution.

unit played vital roles in the properties of the two *n*-type electron acceptors.

**Photovoltaic performance**. The strong absorption at 400–700 nm, as shown in Supplementary Fig. 6a, of the wide-bandgap donor polymer **PM6** is perfectly complementary to the absorption spectra of the two SiO-bridged electron acceptors. In addition, the energy levels of **PM6** ($E_{HOMO} = -5.50$ eV; $E_{LUMO} = -3.67$ eV; estimated from the CV profile of **PM6**, as shown in Supplementary Fig. 6b) also match well with those of the SiO-bridged acceptors (Fig. 5e). Furthermore, the photoluminescence (PL) spectra of the pure **SiOTC** & **SiOTIC** films and the blend films with **PM6** were measured and are shown in Supplementary Fig. 7. **SiOTIC** shows much stronger PL emission than **SiOTC** in the region of 730–1000 nm. Compared with the pure films, both **PM6/SiOTC** and **PM6/SiOTIC** binary blends exhibit complete PL quenching, indicating that efficient photoinduced charge transfer occurs in the films between the donor and acceptor molecules, which is a prerequisite for achieving high photovoltaic performance. Then, we fabricated bulk heterojunction PSCs by employing **PM6** as the donor and **SiOTC** or **SiOTIC** as the acceptor to assess the potential of these SiO-bridged small molecules as promising acceptors with a conventional device structure of ITO/PEDOT: PSS/PM6: acceptors/PNDIT-F3N/Ag. The effects of the regiochemistry and the SiO-synergistic effects on the photovoltaic performance of the two isomeric n-type semiconductors were systematically investigated. The donor/acceptor weight ratio in the devices was optimized to be 1:1.2 in CB solvent with 0.5% DIO additive. The tested current density–voltage ($J - V$) and external quantum efficiency (EQE) curves of the **SiOTC**- and **SiOTIC**-based PSCs are presented in Fig. 5a, b, respectively. The extracted performance parameters are summarized in Table 2. Unfortunately, the PSCs based on as-cast **PM6:SiOTC** showed poor PCEs of 1.73% with very low $J_{sc}$ and FF values. Annealing treatment also did not significantly improve the performance of the **SiOTC**-based device (1.96% PCE). In sharp contrast, we were pleased to find that the isomer **SiOTIC**-based device exhibited a decent PCE of 10.04% upon annealing treatment. The significantly increased performance of the **SiOTIC**-based device was attributed to the simultaneously enhanced $V_{oc}$ (0.92 V) and FF (74.86%) values. From the EQE profiles of these devices, the **SiOTIC**-based device exhibited an elevated photoresponse from 400 to 800 nm with EQE beyond 60% in comparison with that of the **SiOTC**-based device. The different aggregation behaviors of the two isomeric siloxy-bridged acceptors in the blend active layers of the PSCs are responsible for the large difference in photovoltaic performance between the two acceptor-based devices. It is important to note that the **SiOTIC**-based device shows better photovoltaic performance than the reported devices using C-bridged analogs (IDIC)[58, 59] and CO-bridged analogs as the acceptors[61, 63, 64]. As the absorption of **SiOTIC** is complementary to that of **PM6** and **Y6** and the LUMO/HOMO levels of **SiOTIC** (−3.84/−5.57 eV) were also between those of **PM6** (−3.67/−5.50 eV) and **Y6** (−4.10/−5.65 eV), as shown in Supplementary Fig. 8, **SiOTIC** is capable of facilitating electron transfer from **PM6** to **Y6** via cascade charge transfer in devices. Thus, we expanded the application of **SiOTIC** as the third component for ternary PSCs to realize further improvements of the classical efficient photovoltaic system in power conversion efficiency[66]. **SiOTIC** was mixed with a binary blend of **PM6** and **Y6** to fabricate an efficient ternary device. A **PM6:Y6:SiOTIC** weight ratio of 1:1.1:0.1 was employed. Owing to the simultaneous improvement in $V_{oc}$ and FF, the PCE (16.58%) of the ternary device was considerably higher than that of the corresponding binary device based on **PM6:Y6** (15.68%). The $J-V$ and EQE curves of these binary and ternary devices are shown in

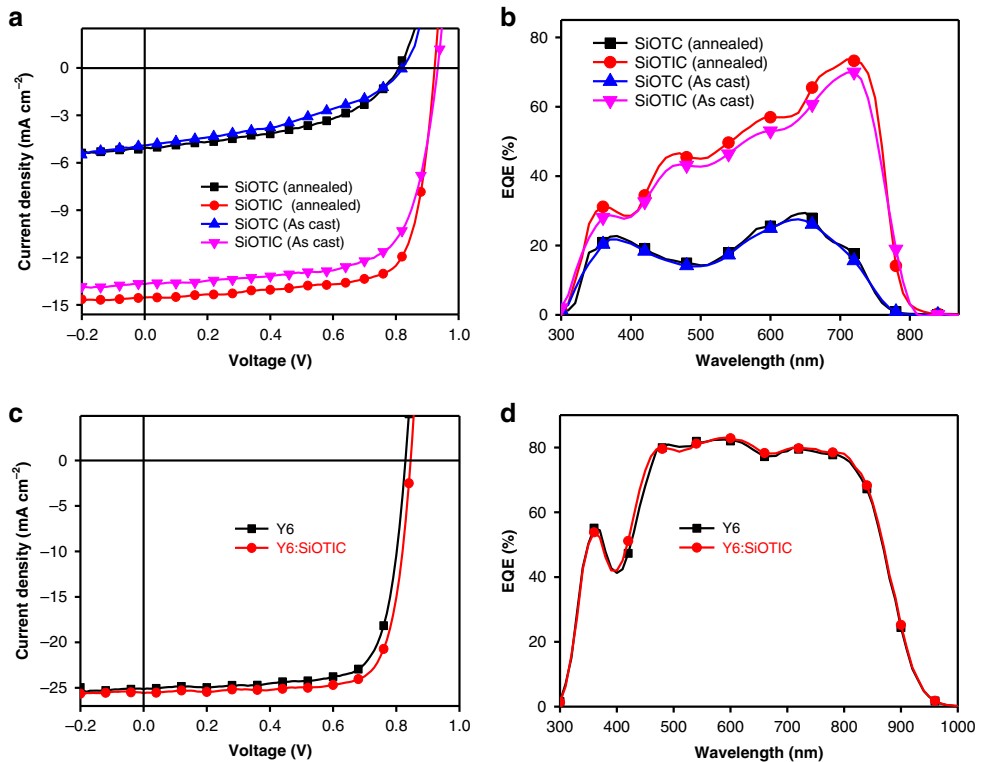

**Fig. 5 Device performance of the corresponding solar cells. a** *J–V* curves of the **PM6:SiOTIC**-based binary device under 100 mW cm$^{-2}$ AM 1.5 G irradiation. **b** The corresponding EQE curves of the **PM6:SiOTIC**-based binary device. **c** J-V curves of the **PM6:Y6** and **PM6:Y6/SiOTIC**-based binary and ternary devices under 100 mW cm$^{-2}$ AM 1.5 G irradiation. **d** EQE curves of the corresponding devices.

**Table 2 The performance parameters of SiOTC- and SiOTIC-based PSCs with PM6 as the donor under 100 mW cm$^{-2}$ AM 1.5 G irradiation$^a$.**

| Acceptors | $V_{oc}$ [V] | $J_{sc}$ [mA cm$^{-2}$] | $J_{Cal}$ [mA cm$^{-2}$] | FF [%] | PCE [%] |
|---|---|---|---|---|---|
| SiOTC (as cast) | 0.82 (0.82 ± 0.002) | 4.92 (4.78 ± 0.13) | 4.74 | 42.85 (41.66 ± 1.16) | 1.73 (1.61 ± 0.11) |
| SiOTC (annealed) | 0.81 (0.81 ± 0.001)$^a$ | 5.06 (4.85 ± 0.21) | 4.91 | 47.77 (47.08 ± 0.68) | 1.96 (1.77 ± 0.18) |
| SiOTIC (as cast) | 0.93 (0.93 ± 0.002) | 13.68 (13.40 ± 0.28) | 13.06 | 69.44 (68.16 ± 1.26) | 8.86 (8.40 ± 0.44) |
| SiOTIC (annealed) | 0.92 (0.92 ± 0.002) | 14.52 (14.15 ± 0.35) | 13.88 | 74.86 (73.83 ± 1.12) | 10.04 (9.67 ± 0.35) |
| Y6 | 0.83 (0.83 ± 0.001) | 25.14 (24.79 ± 0.34) | 24.67 | 75.15 (73.86 ± 1.28) | 15.68 (15.27 ± 0.40) |
| Y6: SiOTIC (1.1:0.1) | 0.85 (0.85 ± 0.001) | 25.29 (24.94 ± 0.33) | 24.91 | 77.11 (75.87 ± 1.23) | 16.58 (16.29 ± 0.38) |

$^a$Average value ± standard deviation were calculated from 10 independent devices.

Fig. 5c, d, and the corresponding photovoltaic parameters are also presented in Table 2. It is worth noting that the determined hole and electron mobilities of binary and ternary blends (Supplementary Fig. 9 and Supplementary Table 1) are in accordance with the different device performances. Incorporation of **SiOTIC** as a guest acceptor significantly enhanced the electron mobility, leading to more balanced carrier mobility in contrast to the **PM6/Y6** binary blend, which is responsible for the improvement in the performance of ternary devices. These results indeed indicate that cooperation between silicon and oxygen in π-systems is a promising strategy for improving the photovoltaic properties of acceptor materials.

To understand the "isomeric effects" on charge transport and recombination in the binary blend, the hole and electron mobility and $J_{sc}$ versus illumination were also measured, as shown in Supplementary Figs. 10 and 11. The mobility calculated by means of the SCLC method is shown in Table S2. The hole mobility of the blend film was $8.5 \times 10^{-6}$ and $1.7 \times 10^{-4}$ cm$^2$ V$^{-1}$ s$^{-1}$ for **SiOTC** and **SiOTIC**, respectively. Furthermore, the electron mobility of the blend film was $5.3 \times 10^{-6}$ and $1.3 \times 10^{-4}$ cm$^2$ V$^{-1}$ s$^{-1}$ for

**SiOTC** and **SiOTIC**, respectively. The extremely low electron and hole mobility of the blend film with **SiOTC** indicates that a loose molecular conformation of **SiOTC** might lead to complicated stacking and diversified aggregation modes, which are harmful to carrier transport. In contrast, the blend film with **SiOTIC** had much higher electron and hole mobility, and the balanced hole and electron transport of **SiOTIC** was also better than that of the other isomer-based blend films, implying that a more compact alkyl chain is of great benefit to suppression of the over-stacking, results in more ordered aggregation and ensures the carrier transfer. The results were also supported by the carrier recombination of the devices, and the slopes from Supplementary Fig. 11 for **SiOTC** and **SiOTIC** were 0.90 and 0.97, respectively. To further study the effect of structure variation on device performance, transient photovoltage (TPV) and transient photocurrent (TPC) measurements were conducted to investigate the carrier lifetime and sweeping out time, as shown in Supplementary Fig. 12 and Supplementary Table 3. The carrier lifetime (τ) was determined by fitting to be 0.60 and 0.74 μs for **SiOTC**- and **SiOTIC**-based devices, respectively.

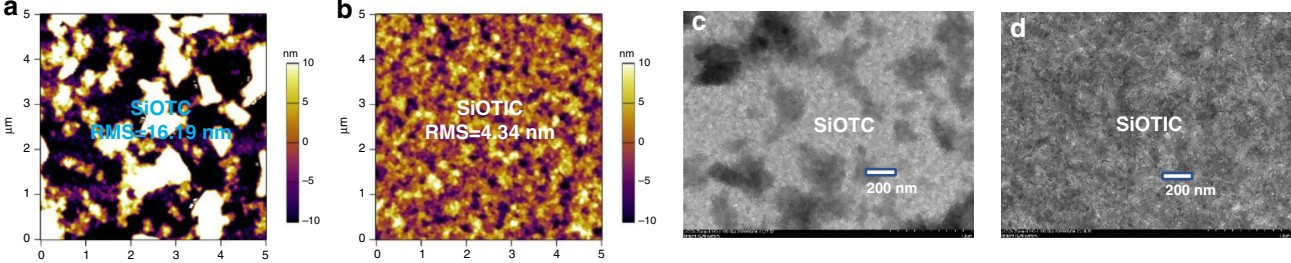

**Fig. 6 AFM and TEM images of the PM6:SiOTC and PM6:SiOTIC blend films. a** AFM images (5 μm × 15 μm) of the **PM6**: **SiOTC** blend film. **b** AFM images (5 μm × 15 μm) of the **PM6**:**SiOTIC** blend film. **c** TEM image of the **PM6**: **SiOTC** blend film. **d** TEM image of the **PM6**:**SiOTIC** blend films.

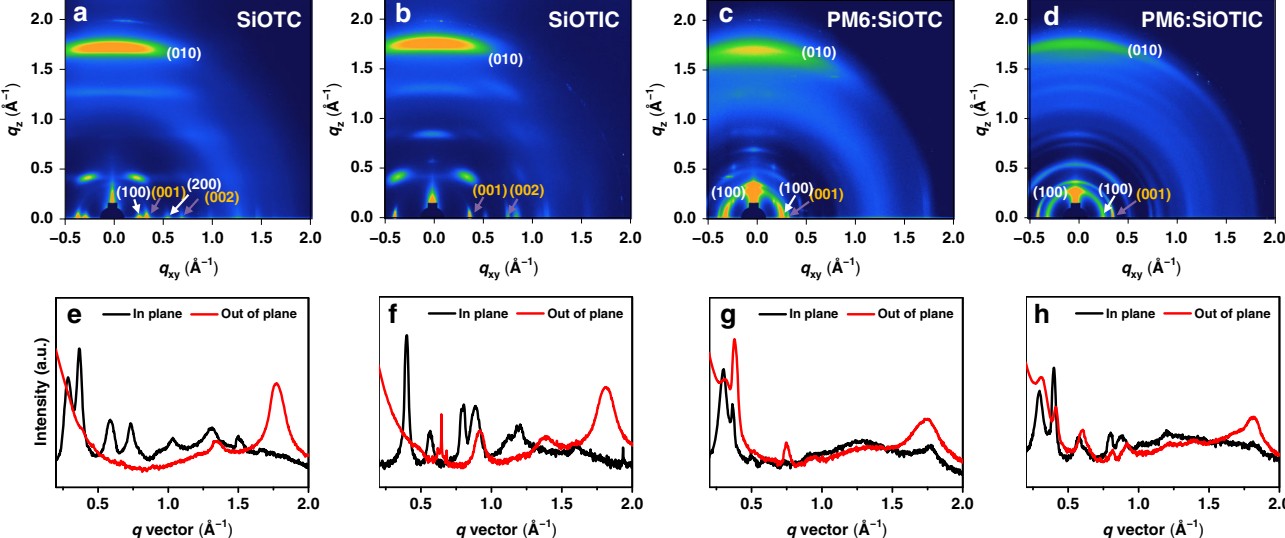

**Fig. 7 GIWAXS results. a** GIWAXS pattern of the **SiOTC** film. **b** GIWAXS pattern of the **SiOTIC** film. **c** GIWAXS pattern of **the PM6:SiOTC** blend film. **d** GIWAXS pattern of the **PM6:SiOTIC** film. **e** The line-cuts of in-plane and out-of-plane **SiOTC**. **f** The line-cuts of in-plane and out-of-plane **SiOTIC**. **g** Line cuts of the in-plane and out-of-plane **PM6:SiOTC** blend films. **h** Line cuts of the in-plane and out-of-plane **PM6:SiOTIC** blend films.

The longer lifetime indicated less recombination, which is consistent with the above description. The sweeping out time ($t_s$) was calculated from the ternary device curves as 0.44 and 0.37 μs for **SiOTC**- and **SiOTIC**-based devices, respectively. The faster $t_s$ for the **SiOTIC**-based device exhibited more efficient charge extraction around electrodes, consistent with the higher $J_{sc}$ and FF values for the device.

**Morphological characterization**. To study the influences of phase-separated morphology on the performance of devices, atomic force microscopy (AFM) and transmission electron microscopy (TEM) were conducted to study the film surface of the blend film with these two acceptors. As shown in Fig. 6a, b (AFM), the **SiOTC**-based film displayed a very rough surface with a root-mean square roughness (RMS) value of 16.19 nm. In contrast, the **SiOTIC** film had a smooth surface morphology with an RMS value of 4.34 nm. The excessive aggregation that appeared in the **SiOTC**-based film led to a large domain size that hindered charge separation. From the TEM images (Fig. 6c, d), the film with **SiOTC** had obvious clusters among the fibers, and the **SiOTIC** film had a dispersive fibrillar nanostructure, which is beneficial for producing efficient exciton separation. These results showed that fine modulation of the molecular structure was crucial for the desired morphology in the solid-state and influenced the optimized performance of PSCs.

Grazing-incidence wide-angle X-ray scattering (GIWAXS) measurements were also carried out to investigate the molecular packing and crystallinity in the neat and blend films. The two-dimensional (2D) GIWAXS patterns are shown in Fig. 7a–d, and the intensity profiles in the out-of-plane (OOP) and in-plane (IP) directions are plotted in Fig. 7e–h. The peak positions, d-spacing, and coherence lengths (CLs) are also presented in Supplementary Table 4. Both **SiOTC** and **SiOTIC** in neat films exhibit obvious π − π (010) stacking peaks in the OOP direction at $q_z = 1.774$ Å$^{-1}$ (**SiOTC**) and 1.813 Å$^{-1}$ (**SiOTIC**), indicating that such molecules are highly ordered with preferential "face-on" orientation. Shorter OOP π − π stacking distances ($d_\pi = 3.466$ Å in **SiOTIC** & $d_\pi = 3.541$ Å in **SiOTC**) were observed in **SiOTIC**, implying that **SiOTIC** possesses more compact π − π stacking, which is beneficial to intermolecular electron transport in films. However, the pure **SiOTC** film exhibits higher crystallinity than **SiOTIC** with a dominant face-on orientation, as reflected by the longer CLs of the OOP (010) peaks (69.158 Å in **SiOTC** and 58.714 Å in **SiOTIC**). Notably, both **SiOTC** and **SiOTIC** display (001) reflections at 0.366 Å$^{-1}$ (**SiOTC**) and 0.398 Å$^{-1}$ (**SiOTIC**) in the IP direction, originating from backbone ordering, which would facilitate intermolecular electron transport.

According to the scattering profiles of blend films, the **PM6/SiOTIC** blend film presented closer π−π stacking ($q_z = 1.815$ Å$^{-1}$, $d = 3.461$ Å in the **PM6/SiOTIC** blend; $q_z = 1.760$ Å$^{-1}$, $d = 3.569$ Å in the **PM6/SiOTC** blend), higher crystalline characteristics

(CLs: 70.139 Å in the **PM6/SiOTIC** blend and 53.058 Å in the **PM6/SiOTC** blend) in the OOP direction and stronger IP backbone (001) stacking ($q_{xy} = 0.398$ Å$^{-1}$, $d = 15.777$ Å and $CL = 279.564$ Å in the **PM6/SiOTIC** blend; $q_{xy} = 0.365$ Å$^{-1}$, $d = 17.234$ Å and $CL = 223.717$ Å) in the **PM6/SiOTC** blend than in the **PM6/SiOTIC** blend, which facilitates charge transport in the vertical direction and suppresses charge carrier recombination, as evidenced by the improved fill factor, charge mobility and $J_{sc}$ in the **SiOTIC**-based device.

## Discussion

In summary, two isomeric SiO-bridged ladder-type π-skeletons (**SiO5Ts**) were designed and synthesized for the first time by using two different tin-free synthetic pathways, showing the advantage of fewer synthetic steps over conventional aromatic fused-rings. In addition, compared with their CO-bridged analogs, the SiO-bridged ladder-type skeletons possess enhanced backbone planarity and lower-lying energy levels of both the LUMO and HOMO. To explore its potential application in organic electronics, two *n*-type A-D-A structured organic semiconductors were constructed by the use of the two isomeric skeletons as the central fused ring D-units. The isomeric effects and the silicon and oxygen synergistic effects of the two SiO-bridged acceptors on the physicochemical and photovoltaic performance were systematically investigated. The results revealed that the molecular geometries, photophysical properties, and aggregation behaviors of the acceptors can be finely adjusted via alteration of the silicon atom positions on the fused-ring central units, thus resulting in a significant effect on the blend film morphology, charge mobility, and photovoltaic performance. As a result, the PSCs based on **SiOTIC** delivered a decent PCE of up to 10.04%, while that of the **SiOTC**-based device only reached 1.96% PCE. Higher device performance can be expected with further efforts on device engineering, such as using appropriate donors, interfacial interlayers, and morphology control. The promising PCE of up to 10.04% given by our first few trials is higher than those of the 5-membered IDT and 6-membered CO5T analog-based PSCs reported in the literature, unambiguously confirming the considerable potential of the SiO-bridged ladder-type π-skeletons for further development of new high-performance semiconductor materials. The structure–property relationship is beneficial to the discovery of new n-type organic semiconductor acceptors for high-performance PSCs.

**Reporting summary**. Further information on research design is available in the Nature Research Reporting Summary linked to this article.

## Data availability

The authors declare that the data supporting the findings of this study are available within the article and its Supplementary Information Files as well as from the corresponding authors on request.

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

## Acknowledgments

D.Z. is grateful for the financial support from the National Natural Science Foundation of China (22022103, 22071114, 21871146), the National Key Research and Development Program of China (2019YFA0210500), the 1000-Talent Youth Program (020/BF180181), the Natural Science Foundation of Tianjin (18JCYBJC20400), and the Fundamental Research Funds for the Central Universities and Nankai University. F.H. is grateful for support from the Shenzhen Nobel Prize Scientists Laboratory Project (C17213101), Guangdong Innovative and Entrepreneurial Research Team Program (2016ZT06G587) and Shenzhen Sci-Tech Fund (KYTDPT20181011104007). We thank the SUSTech Core Research Facilities for the AFM and TEM measurements. Z.-G.Z. is grateful for the financial support from the National Natural Science Foundation of China (51722308).

## Author contributions

Y.Q. and Y.Z. (Yue Zhou) performed chemical synthesis and property characterization of these SiOT5s. H.C. and J.Y. designed and fabricated PSC devices. Y.Z. (Yulin Zhu) conducted the AFM and TEM experiments. Y.C. and C.Y. performed the GIWAXS measurements. C.-W.J. and B.Q. carried out the DFT calculation. D.Z., F.H., Z.-G.Z., and Y.L. developed the concept, designed the experiments, supervised the project and wrote the manuscript.

## Competing interests

The authors declare no competing interests.
