## [Peer Review File · Nature Communications]

Reviewer #1 (Remarks to the Author):

This is a very original and potentially very useful piece of work describing the synthesis of new conjugated cores. In particular, such a synthetic work can be very important for the development of a large number of new n-type semiconductors, often called NFAs. The synthetic approach is appealing and follows new developments for greener materials for greener sources of energy. Originality and great potential impact are clearly present in this work. Having prepared two isomers is also a great asset.

However, before publication many important points should be answered and clarified.

1) in the NMR spectra, in many cases, additional little (and not so little) peaks are present and indicate the presence of impurities. When it happens in the precursors, it is not too serious but even final products show such additional and unidentified peaks. It would be important to show enlarged NMR spectra with a smooth background. Moreover, mass spectra do not certify the purity of the compounds but only indicate that the desired compounds are present. Clear evidences of purity should be given since impurities can affect the performance of the devices. Moreover, to be useful for other researchers, clear and well described purification protocols leading to clean and pure compounds must be reported.

2) Minor points: this reviewer is not convinced of the synergistic properties of having both O and Si atoms in the reported molecules. Synergy means a clear improvement of something compared to a simple addition of both components. Not obvious here.

3) this reviewer is also not convinced that the larger atomic radius of Si is an advantage for the foreseen applications. A longer bond length is clearly useful since it puts further the side chains and that is the main explanation for the so-called ZZ copolymers reported by the authors in the introduction but the larger atomic radius is something different and could only lead to an increase the interchain distance. This reviewer is perhaps wrong on that point but the authors must better explain their argument.

4) many typos are still present in the submitted manuscript and should be carefully removed.

Reviewer #2 (Remarks to the Author):

In this manuscript, the authors have reported two isomeric SiO-bridged ladder-type π -skeletons and used them as the central fused ring D-units for A-D-A type small molecule acceptors. They discussed the effect of isomeric effects and the silicon and oxygen synergistic effects of the two SiO-bridged acceptors on the optical and electronic properties, morphology of films, and performance of OSCs. The OSCs based on PM6:SiOTIC delivered a decent PCE while the PM6:SiOTC showed a poor PCE. However, there are still a variety of issues which need to be addressed.

1. Although the authors synthesized two pentacyclic siloxybridged π -conjugated isomers, SiO5T-5 and SiO5T-10, for the first time, the chemical structure of the π -skeletons lack novelty sufficiently. The CO-bridged analogues have been widely reported and applied.

2. As a number of fused-ring electron acceptors based on A-D-A structure have exhibited high PCE of over 14% (Chem. Soc. Rev., 2020, 49, 2828-2842), the device performance based on PM6:SiOTC and PM6:SiOTIC are relatively low, which is not enough to be published on this stage.

3. The OSCs based on PM6:SiOTIC and PM6:SiOTC exhibited different performance, which were ascribed to the different blend morphology of the two systems. The film morphology is an important point of this work but the morphology is not thoroughly studied and discussed. It would be helpful to add resonant soft X-ray scattering (R-SoXS) and photoluminescence (PL) quenching to support

morphology characterization of active layer.

4. Why a small number of SiOTIC can improve performance of PM6:Y6-based device, the authors need to give the reason.

5. According to TGA, the two isomers exhibited very different thermal decomposition temperatures (234 °C vs 323 °C). The reason of different thermal stability should be given.

Considering the above issues in this paper, at this stage I do not suggest to consider the manuscript for publication.

Reviewer #3 (Remarks to the Author):

In this manuscript, the authors synthesized two isomers, SiOTC and SiOTIC, as non-fullerene acceptors in BHJ-OSC devices. PM6:SiOTC-based device exhibits a PCE of 1.96% and PM6:SiOTIC-based device exhibits a PCE of 10.04%. The photovoltaic performance of binary BHJ-OSC device is too low and is not suitable for publication in prestigious journal of Nature Communication. Moreover, SiO-bridged SiOTIC (10.04%) does not exhibit better photovoltaic performance than CO-bridged TPTIC (10.42%) does (J. Mater. Chem. A, 6, 15933-15941 (2018)). This significantly reduces the novelty of SiO-bridge strategy because there is no improvement in photovoltaic performance. Indeed, introducing SiOTIC as a third component into high-performance PM6:Y6-based device elevates the PCE from 15.62% to 16.58%. However, the authors did not discuss about this PCE increment with plausible statement or characterization. Also, some data of this work is not reliable enough to support the relevant discussion. From the device physics point of view, readers could not obtain sufficient and useful insights from this work. Due to the abovementioned reasons, I do not recommend this manuscript for publication in Nature Communication.

Some weak points of this manuscript are listed as follows:

1. The difference between Egopt and EgCV is too high. I recommend the authors to check the data.
2. The HOMO/LUMO of PM6 provided in this work is not consistent with that in most of the other studies. (

J. Mater. Chem. A, 5, 22180-22185 (2017); J. Mater. Chem. A, 7, 21432-21437 (2019); Angew. Chem. Int. Ed. 59, 9004-9010 (2020)) Please provide the CV diagram of PM6 used in this work.

3. The authors noted the HOMO/LUMO of calculated from DFT simulation. CO-bridged CO5T-5 shows higher HOMO/LUMO than SiO-bridged SiO5T-5 does. Same result can be found in CO5T-10 and SiO5T-10. However, when connecting the same end groups on both flanks of core, CO-bridged TPTIC shows deeper HOMO/LUMO than SiO-bridged SiOTC does. Same result can be found in TPTIC and SiOTIC. Please provide plausible explanation about this.

4. The authors noted the PM6 exhibits strong absorption in 400-700 nm. As a result, PM6 should afford larger bandgap than SiOTC and SiOTIC. However, SiOTC and SiOTIC possess larger bandgap than PM6 does as illustrated in Figure 4 (e). Please correct this mistake.

5. Please discuss the role of SiOTIC in PM6:Y6-based ternary device and note the reason that improves the PCE.

6. Please add the unit of extinction coefficient in Figure 4 (b).

7. Please provide the scale of TEM images in Figure 6.

8. I suggest the authors to proof the manuscript carefully. Many typos, inconsistent format and grammar mistakes can be found in this manuscript. ("implied" to "implied")

9. The authors noted "SiOTIC shows higher melting temperature (289.5 °C) than that of SiOTC (276.3 °C), indicating that it is harder to break the crystallite structure of the SiOTIC in device fabrication. It implied that the thermal annealing might be needed for improving the morphology of the blend

films to ensure the high performance.” Please provide more discussion about these statements logically. Also, the authors did not mention about the thermal annealing condition.

10. The authors noted that “Both of the two isomers films presented broader absorption ranges (500-800 nm) and the red-shifted maximum absorption peaks (699 nm for SiOTC and 717 nm for SiOTIC), indicating that the films of the two isomers showed stronger π - π stacking interactions and more ordered aggregation.” UV-vis absorption is a rough approach to conclude the packing characteristic because the peak width is easily influenced by the thickness and uniformity of thin film. Also, reasons of the relation between absorption and ordered aggregation should be provided. I strongly suggest the authors to measure the corresponding GIWAXS profiles to validate these statements.

In response to the suggestion from Reviewer 1:

In general, we thank this reviewer for acknowledging the originality and great potential of our work and pointing out “the synthesis of new conjugated cores”, which is the key achievement of this work.

Suggestion: In the NMR spectra, in many cases, additional little (and not so little) peaks are present and indicate the presence of impurities. When it happens in the precursors, it is not too serious but even final products show such additional and unidentified peaks. It would be important to show enlarged NMR spectra with a smooth background. Moreover, to be useful for other researchers, clear and well described purification protocols leading to clean and pure compounds must be reported.

Our response: We appreciated this reviewer’s kind suggestion. We have further purified most of the key compounds such as **SiO5T-5**, **11a**, **11b**, **SiOTC** and **SiOTIC** and provided new ¹H and ¹³C NMR spectra with enlarged area in revised SI attached, which are much better than those presented in the original version. We have also provided more details on the synthetic and purification procedures for these new conjugated cores in the revised SI. We assured that these procedures presented in SI for the synthesis of **SiOTC** and **SiOTIC** are highly reproducible.

Suggestion: Minor points: this reviewer is not convinced of the synergistic properties of having both O and Si atoms in the reported molecules. Synergy means a clear improvement of something compared to a simple addition of both components. Not obvious here.

Our response: We appreciate this reviewer’s thoughtful comments. In our manuscript, we use “Synergy” to emphasize the difference and the similarity among SiO-bridge, Si-bridge and O-bridge in π -system. Introduction of Si and O-atom into the π -system at the same time led to some different and unique properties, which are difficult to achieve in Si-bridged or O-bridged π -units. We called it “synergistic effects of both O and Si atoms”.

Suggestion: This reviewer is also not convinced that the larger atomic radius of Si is an advantage for the foreseen applications. The larger atomic radius is something different and could only lead to an increase the interchain distance. This reviewer is perhaps wrong on that point but the authors must better explain their argument.

Our response: We appreciate this reviewer’s thoughtful suggestions. We agree that it is hard to say the larger atomic radius of Si is an advantage or disadvantage for development of new π -units. Indeed, the larger atomic radius of Si seems not to bring some specific advantages for our materials in our manuscript. However, in general, the silicon-containing bonds are easily polarized and possess low bond energy because of the larger atomic radius of Si. These properties make it easier to introduce specific functional groups on silicon atom for tuning the carrier injection and transporting properties in silicon-containing π -units.

Suggestion: Many typos are still present in the submitted manuscript and should be carefully removed.

Our response: Many thanks for the reviewer's careful checking. We have double checked the manuscript and corrected some stylistic and orthographic errors, which we hope meet with the referee's approval.

In response to the suggestions from the Reviewer 2:

Comment: Although the authors synthesized two pentacyclic siloxybridged π -conjugated isomers, SiO5T-5 and SiO5T-10, for the first time, the chemical structure of the π -skeletons lack novelty sufficiently. The CO-bridged analogues have been widely reported and applied.

Our response: We thank the reviewer for his/her assessment. Indeed, the CO-bridged analogues have been widely reported and applied. However, we think that the studies on CO-bridged analogues did not hurt the originality on development of silicon-bridged cores, but inspired us to investigate the effect on incorporation of silicon into the π -fused framework. As mentioned in manuscript, even carbon and silicon have the same electronic configuration of valence electrons, their chemistries are very different. A lot of studies revealed that incorporation of silicon atoms into materials would lead to significant changes on properties in materials. In fact, the elements of group 14 show a greater range of chemical behavior than any other family in the periodic table. The studies on the elements of group 14 have been highly appreciated by the community.

“Carbon/Silicon Switch” as the spark that drives the innovation of molecular structures has been well recognized by material chemists for discovery of new π -materials possessing superb photophysical and electronic properties, which are difficult to achieve in carbon-based π -units in numerous literatures. The longer C–Si bond length (C–Si vs. C–C: ca. 1.87 Å vs. ca. 1.53 Å), higher electropositivity (Si vs. C: 1.74 vs. 2.50), bigger atomic radius (Si vs. C: 111 pm vs. 67 pm) and the particular interactions (σ^* - π^* conjugation) between the Si atom and the π -electron system often led to variation of energy level, increased crystallinity, improved packing ability and higher charge mobility in comparison to those carbon-bridged π -conjugated scaffolds, which have been clearly demonstrated by the wide utilization of diverse 5-membered silicon-bridged π -systems such as siloles, dibenzosiloles, dithienosiloles and related compounds in organic electronics. On the other hand, recent studies revealed that insertion of electron-rich oxygen atoms on the bridge of ladder-type units would greatly improve the light-harvesting capability and the electron-donating capability of the unit, thus bringing unique characteristics into those new organic semiconductor materials to enhance the power conversion efficiency of organic photovoltaics. Taking the advantages of silicon and oxygen atom-doping in the design of the high performance organic semiconductors, development of 6-membered silicon-oxygen-bridged (SiO-bridged) π -conjugated skeletons to investigate the synergistic effects of Si and O atoms in organic electronics is highly appealing and would significantly enhance the structural diversity of π -conjugated materials in a wide range of optoelectronic technologies.

Moreover, herein the synthetic route for siloxy-bridged π -conjugated cores is completely different from the reported CO-bridged analogues. The synthetic route reported for CO-bridged

analogues is incapable of producing these new siloxy-bridged π -conjugated cores. As pointed out by reviewer 1, the synthetic approach presented in this manuscript is appealing and follows new developments for greener materials for greener sources of energy.

Comment: As a number of fused-ring electron acceptors based on A-D-A structure have exhibited high PCE of over 14% (*Chem. Soc. Rev.*, 2020, 49, 2828-2842), the device performance based on PM6:SiOTC and PM6:SiOTIC are relatively low, which is not enough to be published on this stage.

Our response: We thank the reviewer for his/her assessment. I'm sure the reviewer would agree with me that progress in organic electronics highly relies on the development of new π -conjugated skeletons. As the review (*Chem. Soc. Rev.*, 2020, 49, 2828-2842) pointed out, development of new active layer materials are still believed to be the main driving force to address all of the issues in OPVs.

Indeed, since 2015, rapid advances have been achieved in ladder-type non-fullerene acceptors. Among these, most of the reported high performance ladder-type electron acceptors (over 14% PCE) are derived from the ladder-type **IDT**, **CO5T** and **BZPT** backbones. In fact, the ladder-type backbones, which are successfully utilized in high performance NFAs, are still relatively limited. Thus we think that pursuit of a "record efficiency" by molecular and device optimization as well as the further development of new ladder-type backbones are of equal importance in OPVs. For example, viewing the development of the "star acceptor" **Y6** by Zou's group, it started from a known ladder-type donor unit **BZTP** (Synthesized by Cheng et al in *Org. Lett.* **2011**, *13*, 5484–5487) and underwent a continuous molecular and device optimization with contently rising performance to finally reach a "record efficiency" (up to 6% PCE in *ACS Appl. Mater. Interfaces* **2017**, *9*, 31985–31992; up to 12% PCE in *Nat. Commun.* **2019** *10*, 570 and over 15% in *Joule* **2019**, *3*, 1140–1151.). In fact, recently a lot of high-quality studies with synthetic and structural novelty but a lack of "record efficiency" on development of new electron acceptors for OPVs have been published on prestigious journals (10.27% PCE in *Nat. Commun.* **2019**, *10*, 2152; 13.46% PCE in *Nat. Commun.* **2019**, *10*, 519; 13.24% PCE in *Nat. Commun.* **2019**, *10*, 3038; 10.26% PCE in *Angew. Chem. Int. Ed.* **2020**, DOI:10.1002/anie.202010856; 8% PCE in *Angew. Chem. Int. Ed.* **2020**, DOI:10.1002/anie.202009272; 8.63% PCE in *Angew. Chem. Int. Ed.* **2020**, DOI:10.1002/anie.202005662)..

Our work in this manuscript emphasizes development of new ladder-type backbones. Even herein we did not achieved a record efficiency by use of our developed new siloxy-bridged π -conjugated cores, we still think that 10% PCE obtained, new ladder donor units, significantly different synthetic approach and the thorough investigations on the molecular packing, crystallinity, device morphology and structure-property relationship in our manuscript would

attract the reader's interest and trigger the further optimization from material chemists. Obviously, there are still plenty of room for further improving the efficiency based on our new siloxy-bridged π -conjugated cores such as backbone modification, side-chain engineering, and end-cap group substitution.

Suggestion: The film morphology is an important point of this work but the morphology is not thoroughly studied and discussed. It would be helpful to add resonant soft X-ray scattering (R-SoXS) and photoluminescence (PL) quenching to support morphology characterization of active layer.

Our response: We appreciate this reviewer's useful suggestion to help us to further improve the quality of our manuscript. Following the reviewer's comment, first we have carried out Grazing incidence wide-angle X-ray scattering (GIWAXS) measurements to investigate the molecular packing in films. The two dimensional (2D) GIWAXS patterns and the corresponding intensity profiles are presented and discussed in our revised version as following:

Grazing-incidence wide-angle X-ray scattering (GIWAXS) measurements were carried out to investigate the molecular packing and crystallinity in the neat and blend films. The two-dimensional (2D) GIWAXS patterns are shown in Figure 7a-d and the intensity profiles in the out-of-plane (OOP) and in-plane (IP) directions are plotted in Figure 7e-h, respectively. The peak positions, d-spacing as well as coherence lengths (CLs) are also presented in Supplementary Table S4. Both of SiOTC and SiOTIC in neat films exhibit obvious π - π (010) stacking peaks in the OOP direction at $q_z = 1.774 \text{ \AA}^{-1}$ (SiOTC) and 1.813 \AA^{-1} (SiOTIC), indicating such molecules are highly ordered with preferential "face-on" orientation. Shorter OOP π - π stacking distances ($d\pi = 3.466 \text{ \AA}$ in SiOTIC & $d\pi = 3.541 \text{ \AA}$ in SiOTC) presented in SiOTIC, implying that SiOTIC possesses more compact π - π stacking, which is beneficial to intermolecular electron transport in films. However, the pure SiOTC film exhibits higher crystallinity compared with the SiOTIC with a dominant face-on orientation, as reflected by the longer CLs of the OOP (010) peaks (69.158 \AA in SiOTC and 58.714 \AA in SiOTIC). Notably, both of SiOTC and SiOTIC display (001) reflection at 0.366 \AA^{-1} (SiOTC) and 0.398 \AA^{-1} (SiOTIC) in the IP direction, originating from the backbone ordering, which would facilitate the intermolecular electron transport.

According to the scattering profiles of blend films, PM6/SiOTIC blend film presented closer π - π stacking ($q_z = 1.815 \text{ \AA}^{-1}$, $d = 3.461 \text{ \AA}$ in PM6/SiOTIC blend; $q_z = 1.760 \text{ \AA}^{-1}$, $d = 3.569 \text{ \AA}$ in PM6/SiOTC blend), higher crystalline characteristics (CLs: 70.139 \AA in PM6/SiOTIC blend and 53.058 \AA in PM6/SiOTC blend) in OOP direction and stronger IP backbone (001) stacking ($q_{xy} = 0.398 \text{ \AA}^{-1}$, $d = 15.777 \text{ \AA}$ and $CL = 279.564 \text{ \AA}$ in PM6/SiOTIC blend; $q_{xy} = 0.365 \text{ \AA}^{-1}$, $d = 17.234 \text{ \AA}$ and $CL = 223.717 \text{ \AA}$) in PM6/SiOTC blend than that of PM6/SiOTIC, which would facilitate charge transport in the vertical direction and suppress charge carrier recombination as

evidenced by the improved fill factor, charge mobility and J_{sc} in SiOTIC-based device.

Figure 7. GIWAXS patterns of SiOTC (a), SiOTIC (b), PM6:SiOTC (c) and PM6:SiOTIC (d) films and the corresponding line-cuts of in-plane and the out-of-plane of SiOTC (e), SiOTIC (f), PM6:SiOTC (g) and PM6:SiOTIC (h) films, respectively.

Moreover, photoluminescence (PL) quenching measurements were also performed and presented as Figure S7 in the revised Supplementary Information. The PL quenching result has been discussed in the revised manuscript as following:

Furthermore, the photoluminescence (PL) spectra of the pure SiOTC & SiOTIC films and the blend films with PM6 were measured and shown in the SI as Figures S7. SiOTIC shows much stronger PL emission than SiOTC in the region of 730–1000 nm. Compared with the pure films, both PM6/SiOTC and PM6/SiOTIC binary blends exhibit a complete PL quenching, indicating that efficient photoinduced charge transfer occurs in the films between the donor and acceptor molecules, which is a prerequisite for achieving high photovoltaic performance.

Figure S7. Photoluminescence (PL) spectra of the pure SiOTC & SiOTIC films and the blend films with PM6.

Suggestion: Why a small number of SiOTIC can improve performance of PM6:Y6-based device, the authors need to give the reason.

Our response: We thank the reviewer's constructive suggestion for further improving this manuscript. As shown in Figure S8 in the revised SI, the absorption of SiOTIC is complementary to PM6 and Y6 and the LUMO/HOMO levels of SiOTIC (-3.84/-5.57 eV) were also posited in between those of PM6 (-3.67/-5.50 eV) and Y6 (-4.10/-5.65 eV). SiOTIC is capable of facilitating electron transfer from PM6 to Y6 via a cascade charge transfer in ternary device. To further clarify the influence of incorporation of SiOTIC guest acceptor on charge transport properties, the mobilities of holes (μ_h) and electrons (μ_e) in the binary and ternary blends were also determined by space-charge limited current (SCLC) measurement as shown in Figure S9 and Table S1 in our revised Supplementary Information. The enhanced electron mobility and more balanced carrier mobility in PM6/SiOTIC/Y6 ternary blend with regard to PM6/Y6 binary blend can rationalize for the improved performance of ternary device. In general, to address the reason, we tried our best to conduct a lot of additional experiments and come out the following comments in the revised version:

As the absorption of SiOTIC is complementary to PM6 and Y6 and the LUMO/HOMO levels of SiOTIC (-3.84/-5.57 eV) were also posited in between those of PM6 (-3.67/-5.50 eV) and Y6 (-4.10/-5.65 eV) as shown in Figure S8, SiOTIC is capable of facilitating electron transfer from

PM6 to Y6 via a cascade charge transfer in devices. Thus we expanded the application of SiOTIC as the third component for ternary PSCs to realize further improvements of the classical efficient photovoltaic system in the power conversion efficiency⁶⁶. SiOTIC was mixed with a binary blend of PM6 and Y6 to fabricate an efficient ternary device. A PM6:Y6:SiOTIC weight ratio of 1:1.1:0.1 was employed. Owing to the simultaneous improvement in the Voc and FF, the PCE (16.58%) of the ternary device is considerably higher than that of the corresponding binary device based on PM6:Y6 (15.68%). The J-V and EQE curves of this binary and ternary devices were shown in Figure 5c and d, and the corresponding photovoltaic parameters were also presented in Table 2. It is worthy to note that the determined hole and electron mobilities of binary and ternary blends (Figure S9 and Table S1 in Supplementary Information) are in accordance with the different device performances. Incorporation of SiOTIC as guest acceptor significantly enhanced the electron mobility, leading to more balanced carrier mobility in contrast to the PM6/Y6 binary blend, which is in response to the improvement of the performance in ternary device. These results indeed indicate that silicon and oxygen cooperation in π -systems is a promising strategy for improving the photovoltaic properties of the acceptor materials.

Figure S8. (a) Normalized absorption spectra of **PM6**, **SiOTIC** and **Y6** in film and (b) Energy level profiles of **PM6**, **SiOTIC** and **Y6**.

Figure S9. The J-V curves of hole-only (a) and electron-only (b) devices based on

binary and ternary blend films.

Table S1. Extracted mobility of the hole and electron-only devices based on binary and ternary blend films.

Device	Hole Mobility ($\text{cm}^2 \text{v}^{-1} \text{s}^{-1}$)	Electron Mobility ($\text{cm}^2 \text{v}^{-1} \text{s}^{-1}$)	Hole/Electron
Binary (PM6/Y6)	3.2×10^{-4}	2.9×10^{-4}	1.4
Ternary (PM6/SiOTIC/Y6)	5.4×10^{-4}	4.5×10^{-4}	1.2

Suggestion: According to TGA, the two isomers exhibited very different thermal decomposition temperatures (234 °C vs 323 °C). The reason of different thermal stability should be given.

Our response: Many thanks for the reviewer's careful checking. To confirm this result, thermal decomposition temperatures of SiOTC and SiOTIC was re-measured by the thermogravimetric analysis (TGA). In fact, we found that the TGA profiles for both of them are quite similar. The difference in the original version attributed to the presence of impurities in SiOTC. We have already updated TGA profiles as Figure S5 in our revised Supplementary Information. The updated TGA profiles is shown as following:

In response to the suggestions from the Reviewer 3:

Suggestion: The photovoltaic performance of binary BHJ-OSC device is too low and is not suitable for publication in prestigious journal of Nature Communication.

Our response: We thank the reviewer for his/her assessment. The main achievements on our study are that we solved challenges on chemical synthesis of siloxy-bridged π -conjugated skeletons and for the first time design and successfully construct two new isomeric pentacyclic siloxy-bridged π -conjugated frameworks. Furthermore, we for the first time demonstrated that incorporation of the Si-O bridge into π -conjugated frameworks significantly enhanced the molecular thermal stability and planarity and offer greater tunability on the molecular energy levels and crystallinity.

Progress in organic electronics highly relies on the development of new π -conjugated skeletons. Pursuit of a “record efficiency” by molecular and device optimization/engineering as well as the further development of new ladder-type backbones are of equal importance in OPVs. Study on the development of new π -conjugated core involving distinct synthetic routes is worthy to be highly appreciated. For example, viewing the development of the “star acceptor” **Y6** by Zou’s group, it started from a known ladder-type donor unit **BZTP** (Synthesized by Cheng et al in *Org. Lett.* **2011**, *13*, 5484–5487) and underwent a continuous molecular and device optimization with continually rising performance to finally reach a “record efficiency” (up to 6% PCE in *ACS Appl. Mater. Interfaces* **2017**, *9*, 31985–31992; up to 12% PCE in *Nat. Commun.* **2019**, *10*, 570 and over 15% in *Joule* **2019**, *3*, 1140–1151.). In fact, recently a lot of high-quality studies with synthetic and structural novelty but a lack of “record efficiency” on development of new electron acceptors for OPVs have been published on prestigious journals (10.27% PCE in *Nat. Commun.* **2019**, *10*, 2152; 13.46% PCE in *Nat. Commun.* **2019**, *10*, 519; 13.24% PCE in *Nat. Commun.* **2019**, *10*, 3038; 10.26% PCE in *Angew. Chem. Int. Ed.* **2020**, DOI:10.1002/anie.202010856; 8% PCE in *Angew. Chem. Int. Ed.* **2020**, DOI:10.1002/anie.202009272; 8.63% PCE in *Angew. Chem. Int. Ed.* **2020**, DOI:10.1002/anie.202005662.). Indeed, since 2015, rapid advances have been achieved in ladder-type non-fullerene acceptors. Among these, most of the reported high performance ladder-type electron acceptors (over 14% PCE) are derived from the ladder-type **IDT**, **CO5T** and **BZPT** backbones possessing similar synthetic routes. In fact, the ladder-type backbones, which have been successfully utilized in high performance NFAs, are still relatively limited. Thus further development of new active layer materials especially for new ladder-type π -backbones are one of the main driving force to address the issues in OPVs.

Our work in this manuscript emphasizes development of new ladder-type backbones. Even herein we did not achieved a record efficiency by use of our developed new siloxy-bridged

π -conjugated cores, we still think that 10% PCE obtained, new ladder donor units, significantly different synthetic approach and the thorough investigations on the molecular packing, crystallinity, device morphology and structure-property relationship in our manuscript clearly demonstrated the great potential of this SiO-bridged ladder-type unit for further development of new high performance semiconductor materials and would certainly attract tremendous attention as well trigger the further optimization from material chemists.

Suggestion: SiO-bridged SiOTIC (10.04%) does not exhibit better photovoltaic performance than CO-bridged TPTIC (10.42%) does (*J. Mater. Chem. A*, 6, 15933-15941 (2018)). This significantly reduces the novelty of SiO-bridge strategy because there is no improvement in photovoltaic performance.

Our response: We thank the reviewer for his/her assessment. As we presented in the manuscript, SiO-bridged materials herein are significant different from CO-bridged materials both in photophysical and electronic properties as well synthetic routes. SiO-bridged materials are not the CO-bridged materials' competitors. Both of them possessed their own unique properties. As proved, different donors are needed to ensure the performance in the CO-bridged TPTIC- (*J. Mater. Chem. A*, 2018, 6, 15933-15941) and SiO-bridged SiOTIC-based OSCs. Our study emphasized the synthetic challenge and the novelty of π -conjugated core as well as the structure-property relationship of Si-O bridged π -conjugated core in OPVs. We care more about new materials rather than a "record efficiency". Viewing from a chemist, "10.04%" is not so different from "10.42%". The comparable performance clearly demonstrated the great potential of this SiO-bridged ladder-type unit in organic electronics.

Suggestion: The difference between E_{g}^{opt} and E_{g}^{CV} is too high. I recommend the authors to check the data.

Our response: We appreciate the reviewer's careful proofreading. E_{g}^{opt} in our manuscript was calculated from the onset wavelength of the molecules in film. In contrast, E_{g}^{CV} was calculated from CV diagram of the corresponding compounds in DCM solution, because we failed to get a clear CV profiles of these compounds in film. We added the corresponding footnote at Table 1 in the revised version.

Suggestion: The HOMO/LUMO of PM6 provided in this work is not consistent with that in most of the other studies. Please provide the CV diagram of PM6 used in this work.

Our response: We appreciate the reviewer's careful proofreading. We have corrected the HOMO/LUMO level of PM6 in the revised version and we also provided the CV and absorption diagram of PM6 used in our work as follow in the revised SI:

Figure S6. Absorption of PM6 (a) and CV profile of the PM6 donor (b).

Suggestion: The authors noted the HOMO/LUMO of calculated from DFT simulation. CO-bridged CO5T-5 shows higher HOMO/LUMO than SiO-bridged SiO5T-5 does. Same result can be found in CO5T-10 and SiO5T-10. However, when connecting the same end groups on both flanks of core, CO-bridged TPTC shows deeper HOMO/LUMO than SiO-bridged SiOTC does. Same result can be found in TPTIC and SiOTIC. Please provide plausible explanation about this.

Our response: We appreciate the reviewer's thoughtful suggestion. The DFT calculation indicates that the difference of energy level between SiOTC/SiOTIC and TPTC/TPTIC originated from the different molecular geometric configuration. In comparison with simple ladder skeletons SiO5T-5/10 and CO5T-5/10, those acceptors based on SiO5T-5/10 or CO5T-5/10 possess push-pull driving forces to facilitate electron delocalization viewing from molecular structure. From DFT calculation, the planarity of the electron donor/acceptor units on SiOTC/SiOTIC is better than those on TPTC/TPTIC as shown in Figure S4 in revised SI. Planarization between adjacent aromatic units allows parallel p-orbital interactions to extend conjugation and facilitate delocalization. This in turn leads to a decrease in the bond length alternation (BLA), enhanced push-pull effect and reduction of the band gap. To provide an explanation, we added some comments and Figures in revised manuscript and SI as following: DFT calculations revealed that the better planarity and enhanced push-pull effect between the electron donor/acceptor units on SiOTC/SiOTIC as shown in Figure S3-4 is responsible for this changes on energy level.

Figure S4. The optimal molecular geometries of **SiOTC** and **SiOTIC** as well as their carbon-analogues (**TPTC** and **TPTIC**) by DFT.

Suggestion: The authors noted the PM6 exhibits strong absorption in 400-700 nm. As a result, PM6 should afford larger bandgap than SiOTC and SiOTIC. However, SiOTC and SiOTIC possess larger bandgap than PM6 does as illustrated in Figure 4 (e). Please correct this mistake.

Our response: We appreciate the reviewer's careful proofreading. We measured the CV, absorption of **PM6** as shown in Figure S6 in the revised SI and the following. Now the energy level and band gap energy of **PM6** in Figure 4e have also been corrected in the revised manuscript.

Figure S6. Absorption of PM6 (a) and CV profile of the PM6 donor (b).

Figure 4e in the revised manuscript

Suggestion: Please discuss the role of SiOTIC in PM6:Y6-based ternary device and note the reason that improves the PCE.

Our response: We appreciate this reviewer's thoughtful suggestion. As shown in Figure S8 in the revised SI, the absorption of SiOTIC is complementary to PM6 and Y6 and the LUMO/HOMO levels of SiOTIC (-3.84/-5.57 eV) were also posited in between those of PM6 (-3.67/-5.50 eV) and Y6 (-4.10/-5.65 eV). These results suggest that SiOTIC is capable of facilitating electron transfer from PM6 to Y6 via a cascade charge transfer in ternary device. To further clarify the influence of incorporation of SiOTIC guest acceptor on charge transport properties, the mobilities of holes (μ_h) and electrons (μ_e) in the binary and ternary blends were also determined by space-charge limited current (SCLC) measurement as shown in Figure S9 and Table S1 in our revised Supplementary Information. We found that incorporation of SiOTIC as guest acceptor significantly enhanced the electron mobility, leading to more balanced carrier mobility in contrast to the PM6/Y6 binary blend, which is in response for the improvement of the PCE. In general, to address the reason, we tried our best to conduct a lot of additional experiments and come out the following comments in the revised version:

As the absorption of SiOTIC is complementary to PM6 and Y6 and the LUMO/HOMO levels of SiOTIC (-3.84/-5.57 eV) were also posited in between those of PM6 (-3.67/-5.50 eV) and Y6 (-4.10/-5.65 eV) as shown in Figure S8, SiOTIC is capable of facilitating electron transfer from PM6 to Y6 via a cascade charge transfer in devices. Thus we expanded the application of SiOTIC as the third component for ternary PSCs to realize further improvements of the classical

efficient photovoltaic system in the power conversion efficiency⁶⁶. SiOTIC was mixed with a binary blend of PM6 and Y6 to fabricate an efficient ternary device. A PM6:Y6:SiOTIC weight ratio of 1:1.1:0.1 was employed. Owing to the simultaneous improvement in the Voc and FF, the PCE (16.58%) of the ternary device is considerably higher than that of the corresponding binary device based on PM6:Y6 (15.68%). The J-V and EQE curves of this binary and ternary devices were shown in Figure 5c and d, and the corresponding photovoltaic parameters were also presented in Table 2. It is worthy to note that the determined hole and electron mobilities of binary and ternary blends (Figure S9 and Table S1 in Supplementary Information) are in accordance with the different device performances. Incorporation of SiOTIC as guest acceptor significantly enhanced the electron mobility, leading to more balanced carrier mobility in contrast to the PM6/Y6 binary blend, which is responsible for the improvement of the performance in ternary device. These results indeed indicate that silicon and oxygen cooperation in π -systems is a promising strategy for improving the photovoltaic properties of the acceptor materials.

Figure S8. (a) Normalized absorption spectra of **PM6**, **SiOTIC** and **Y6** in film and (b) Energy level profiles of **PM6**, **SiOTIC** and **Y6**.

Figure S9. The J-V curves of hole-only (a) and electron-only (b) devices based on binary and ternary blend films.

Table S1. Extracted mobility of the hole and electron-only devices based on binary and ternary blend films.

Device	Hole Mobility ($\text{cm}^2 \text{v}^{-1} \text{s}^{-1}$)	Electron Mobility ($\text{cm}^2 \text{v}^{-1} \text{s}^{-1}$)	Hole/Electron
Binary (PM6/Y6)	3.2×10^{-4}	2.9×10^{-4}	1.4
Ternary (PM6/SiOTIC/Y6)	5.4×10^{-4}	4.5×10^{-4}	1.2

Suggestion: Please add the unit of extinction coefficient in Figure 4 (b).

Our response: We appreciate the reviewer's careful proofreading. We have added the unit of extinction coefficient in Figure 4b in the revised version as shown in following:

Figure 4b in the revised manuscript

Suggestion: Please provide the scale of TEM images in Figure 6.

Our response: We appreciate the reviewer's careful proofreading. We have fixed the TEM images of Figure 6c-d with the scale in the revised version as shown in following:

Figure 6. AFM images ($5 \mu\text{m} \times 15 \mu\text{m}$; a & b) and TEM images (c & d) of the PM6: SiOTC and PM6:SiOTIC blend films.

Suggestion: I suggest the authors to proof the manuscript carefully. Many typos, inconsistent format and grammar mistakes can be found in this manuscript. ("implied" to "implied").

Our response: We appreciate the reviewer's careful proofreading. We have double checked the

manuscript and corrected some stylistic and orthographic errors, which we hope meet with the referee's approval.

Suggestion: The authors noted "SiOTIC shows higher melting temperature (289.5 °C) than that of SiOTC (276.3 °C), indicating that it is harder to break the crystallite structure of the SiOTIC in device fabrication. It implied that the thermal annealing might be needed for improving the morphology of the blend films to ensure the high performance." Please provide more discussion about these statements logically. Also, the authors did not mention about the thermal annealing condition.

Our response: We appreciate this helpful suggestion. We realized that the slightly different melting temperatures can't give us so much useful information about the crystallite structure of these two compounds. Thus, According to the GIWAXS results, we have changed the sentence to "...indicating that the SiOTIC may form more compact stacking in film, which is harder to break it, and the thermal annealing might be needed in device fabrication." In revised manuscript. Moreover, we also provided the thermal annealing condition for device fabrication in the revised Supplementary Information.

Suggestion: Also, reasons of the relation between absorption and ordered aggregation should be provided. I strongly suggest the authors to measure the corresponding GIWAXS profiles to validate these statements.

Our response: We thank the reviewer for this thoughtful suggestion to improve our manuscript. We have carried out Grazing incidence wide-angle X-ray scattering (GIWAXS) measurements to investigate the molecular packing in films. The two dimensional (2D) GIWAXS patterns and the corresponding intensity profiles are presented as Figure 7 and discussed in our revised version as follow:

Grazing-incidence wide-angle X-ray scattering (GIWAXS) measurements were also carried out to investigate the molecular packing and crystallinity in the neat and blend films. The two-dimensional (2D) GIWAXS patterns are shown in Figure 7a-d and the intensity profiles in the out-of-plane (OOP) and in-plane (IP) directions are plotted in Figure 7e-h, respectively. The peak positions, d-spacing as well as coherence lengths ((CLs) are also presented in Supplementary Table S4. Both of SiOTC and SiOTIC in neat films exhibit obvious π - π (010) stacking peaks in the OOP direction at $q_z = 1.774 \text{ \AA}^{-1}$ (SiOTC) and 1.813 \AA^{-1} (SiOTIC), indicating such molecules are highly ordered with preferential "face-on" orientation. Shorter OOP π - π stacking distances ($d_\pi = 3.466 \text{ \AA}$ in SiOTIC & $d_\pi = 3.541 \text{ \AA}$ in SiOTC) presented in SiOTIC, implying that SiOTIC possesses more compact π - π stacking, which is beneficial to intermolecular electron transport in films. However, the pure SiOTC film exhibits higher crystallinity compared with the SiOTIC with a dominant face-on orientation, as reflected by the longer CLs of the OOP (010) peaks (69.158 \AA in SiOTC and 58.714 \AA in SiOTIC). Notably,

both of SiOTC and SiOTIC display (001) reflection at 0.366 \AA^{-1} (SiOTC) and 0.398 \AA^{-1} (SiOTIC) in the IP direction, originating from the backbone ordering, which would facilitate the intermolecular electron transport.

According to the scattering profiles of blend films, PM6/SiOTIC blend film presented closer π - π stacking ($q_z = 1.815 \text{ \AA}^{-1}$, $d = 3.461 \text{ \AA}$ in PM6/SiOTIC blend; $q_z = 1.760 \text{ \AA}^{-1}$, $d = 3.569 \text{ \AA}$ in PM6/SiOTC blend), higher crystalline characteristics (CLs: 70.139 \AA in PM6/SiOTIC blend and 53.058 \AA in PM6/SiOTC blend) in OOP direction and stronger IP backbone (001) stacking ($q_{xy} = 0.398 \text{ \AA}^{-1}$, $d = 15.777 \text{ \AA}$ and CL = 279.564 \AA in PM6/SiOTIC blend; $q_{xy} = 0.365 \text{ \AA}^{-1}$, $d = 17.234 \text{ \AA}$ and CL = 223.717 \AA) in PM6/SiOTC blend than that of PM6/SiOTIC, which would facilitate charge transport in the vertical direction and suppress charge carrier recombination as evidenced by the improved fill factor, charge mobility and J_{sc} in SiOTIC-based device.

Figure 7. GIWAXS patterns of SiOTC (a), SiOTIC (b), PM6:SiOTC (c) and PM6:SiOTIC (d) films and the corresponding line-cuts of in-plane and the out-of-plane of SiOTC (e), SiOTIC (f), PM6:SiOTC (g) and PM6:SiOTIC (h) films, respectively.

Reviewer #1 (Remarks to the Author):

Although the authors argued more than they changed their manuscript, I am pleased with their explanations and therefore, this manuscript can be published without further modification.

Reviewer #2 (Remarks to the Author):

The authors successfully replied to all my comments. Therefore, I recommend the paper to be published as is.

Reviewer #3 (Remarks to the Author):

The authors' effort in this revised manuscript indeed explains some unclear result and statement. However, the core reason I do not recommend this manuscript for publication is its insufficient novelty. As I mentioned before, there is no improvement in comparison with the authors' previous publication, *J. Mater. Chem. A*, 6, 15933-15941 (2018). Also, the authors add Figure S6 to check the optical and electrochemical properties of PM6. However, the normalized solution absorption of PM6 starts from 0.2, which can be attributed to its base line problem in measurement. Moreover, the onset of reduction potential of PM6 can not be obtained by the cut lines as shown in Figure S6(b). Owing to the abovementioned problem, I still refuse to recommend this manuscript for publication in this prestigious journal of *Nature Communication*.

Response to the comments and suggestions from Reviewer #3:

Comment: The authors' effort in this revised manuscript indeed explains some unclear result and statement. However, the core reason I do not recommend this manuscript for publication is its insufficient novelty. As I mentioned before, there is no improvement in comparison with the authors' previous publication, *J. Mater. Chem. A*, 6, 15933-15941 (2018).

Our response: We thank the reviewer for his/her assessment. We still think that the studies on CO-bridged analogues did not hurt the originality on development of silicon-bridged cores, but inspired us to investigate the effect on incorporation of silicon into the π -fused framework. As mentioned in manuscript, even carbon and silicon have the same electronic configuration of valence electrons, their chemistries are very different. A lot of studies revealed that incorporation of silicon atoms into materials would lead to significant changes on properties in materials. "Carbon/Silicon Switch" as the spark that drives the innovation of molecular structures has been well recognized by material chemists for discovery of new π -materials possessing superb photophysical and electronic properties, which are difficult to achieve in carbon-based π -units in numerous literatures. As we presented in the manuscript, SiO-bridged materials herein are significant different from CO-bridged materials both in photophysical and electronic properties as well synthetic routes. ***SiO-bridged materials are not the CO-bridged materials' competitors.*** Both of them possessed their own unique properties. As proved, herein the synthetic route for siloxy-bridged π -conjugated cores is completely different from the reported CO-bridged analogues. The synthetic route reported for CO-bridged analogues is incapable of producing these new siloxy-bridged π -conjugated cores. Moreover, different donors are needed to ensure the performance in the CO-bridged **TPTIC-** (*J. Mater. Chem. A*, 2018, 6, 15933-15941) and SiO-bridged **SiOTIC-**based OSCs.

In general, our study emphasized the synthetic challenge and the novelty of π -conjugated core as well as the structure-property relationship of Si-O bridged π -conjugated core in OPVs. We care more about new materials rather than a "record efficiency". Viewing from a chemist,

“10.04%” is not so different from “10.42%”. The comparable performance clearly demonstrated the great potential of this SiO-bridged ladder-type unit in organic electronics.

Comment: the authors add Figure S6 to check the optical and electrochemical properties of PM6. However, the normalized solution absorption of PM6 starts from 0.2, which can be attributed to its base line problem in measurement. Moreover, the onset of reduction potential of PM6 can not be obtained by the cut lines as shown in Figure S6(b).

Our response: Many thanks for the reviewer’s careful checking. We re-measured the absorption spectra and CV of PM6 polymer. The new absorption and CV profiles on PM6 polymer have been presented as Supplementary Figure 6 in the revised SI.

Supplementary Figure 6. Absorption (a) and CV profile of the PM6 donor (b).